# Comparative analysis of shared and unique mechanisms important for diverse strains of *Pasteurella multocida* to cause systemic infection in mice

**Thomas R. Smallman**[1,2], **Xiaochu Wang**[1,2], **Xenia Kostoulias**[1,2,3], **Amy Wright**[1,2], **Faye C. Morris**[1,2], **Marina Harper**[1,2‡], **John D. Boyce**[1,2‡]*

1 Infection Program, Monash Biomedicine Discovery Institute and Department of Microbiology, Monash University, Melbourne, Victoria, Australia, 2 Centre to Impact AMR, Monash University, Melbourne, Victoria, Australia, 3 Department of Infectious Diseases, Alfred Health and School of Translational Medicine, Monash University, Melbourne, Victoria, Australia

‡ These authors are Joint senior authors.
* john.boyce@monash.edu

**Editor:** Sebastian Suerbaum, LMU Munich Max von Pettenkofer Institute of Hygiene and Medical Microbiology: Ludwig-Maximilians-Universitat Munchen Max von Pettenkofer-Institut fur Hygiene und Medizinische Mikrobiologie, GERMANY

## Abstract

*Pasteurella multocida* is a Gram-negative bacterium that causes a range of distinct diseases in livestock animals. Different *P. multocida* diseases are associated with different capsule and lipopolysaccharide (LPS) types, but little else is known about what underpins this disease specificity. In this study, we utilised transposon-directed insertion site sequencing (TraDIS) to identify genes required for growth in rich media, and genes important for survival during systemic infections in BALB/c mice, for two diverse *P. multocida* strains; VP161 (capsule type A and LPS type L1) and M1404 (capsule type B and LPS type L2). Analysis of growth in heart infusion broth showed that both VP161 and M1404 shared 461 genes essential for growth in rich media, with 95% of these rich media-essential genes present in all publicly available closed *P. multocida* genomes. *In vivo* fitness analysis identified 63 and 94 genes important for VP161 and M1404 survival in BALB/c mice, respectively. Only 35 homologs were identified as important for survival in both strains, showing that conserved biological systems can be differentially important for different *P. multocida* strains. Investigation of proteins involved in the catabolite response showed that an active cyclic-adenosine monophosphate (cAMP) receptor protein (CRP) was required for maximal fitness in M1404. Furthermore, disrupting CRP or cAMP production also reduced capsule production in M1404, but increased capsule production in VP161, demonstrating that these *P. multocida* strains have different regulatory systems for crucial virulence factors.

**Data availability statement:** The M1404 genome was submitted to Genbank with the biosample accession number SAMN49532731. The TraDIS data analysis output files have been uploaded to the NCBI Gene Expression Omnibus and are available under accession number GSE300598.

**Funding:** This work was supported by the Australian Research Council Discovery Project DP210103610 to JB and MH (https://www.arc.gov.au). The funders had no role in study design, data collection and analysis, decision to publish, or preparation of the manuscript.

**Competing interests:** The authors have declared that no competing interests exist.

## Author summary

*Pasteurella multocida* is an important livestock pathogen, causing several distinct severe diseases in many different livestock animals. *P. multocida* can spread rapidly throughout animal populations, with peracute infections causing death within 48 h, resulting in large outbreaks with high mortality. Host predilection and disease presentation often correlate with the capsule and lipopolysaccharide type produced by the causative strain. However, the processes that allow certain strains to cause a particular disease are not well understood. In this study, we have comprehensively identified genes required for two *P. multocida* strains to cause systemic infection in mice, showing that these diverse strains have differential requirements for survival during a systemic infection. This information is crucial for understanding *P. multocida* diseases and for the development of new strategies to combat infection.

## Introduction

*Pasteurella multocida* is an upper respiratory tract commensal of many mammal and bird species [1–3], but the bacterium can also cause serious disease, including fowl cholera in birds, haemorrhagic septicaemia in ungulates, atrophic rhinitis in pigs, and pneumonia in most livestock animals [4]. Strains of *P. multocida* are commonly classified based on capsule type (namely A, B, D, E and F) and/or lipopolysaccharide (LPS) genotype (namely L1-L9) [5–7]. *P. multocida* disease presentation and host predilection often correlates with genotype of surface structures produced by the infecting strain [4]. For instance, fowl cholera is caused primarily by strains with capsule type A or F, and LPS type L1, L3 or L6 [5,8–10], while haemorrhagic septicaemia is only caused by strains with capsule type B or E, and LPS type L2 or L5 [11–13]. Apart from capsule and LPS type, very little is known about the pathogenic mechanisms that are unique to strains causing these different diseases or what pathogenic mechanisms are shared by such strains.

To cause disease, *P. multocida* often colonises the upper respiratory tract before spreading to the lower respiratory tract and can rapidly enter the blood stream via tissue damage or the lymphatic system [14–16]. Entry directly into the bloodstream via skin has also been observed [4]. During a natural systemic infection *P. multocida* colonises several different tissues [4,14,16]. *P. multocida* disease often progresses rapidly and in peracute cases death can occur within 48 h of infection [17,18]. Mouse models of *P. multocida* systemic infection are initiated by intraperitoneal injection [19,20]. In such mouse models, *P. multocida* rapidly enters the bloodstream and colonises several tissues, mimicking natural systemic infections [19,20]. *P. multocida* is readily transmitted between animals within a population, either directly or via environmental contamination. Large *P. multocida* disease outbreaks occur in both wild and livestock animal populations [21–25], with increased susceptibility linked to environmental stresses such as increased temperature and humidity, as well as animal

crowding [26]. Outbreaks of *P. multocida* disease in livestock result in high mortality and large economic losses. Vaccination is an important preventative measure against *P. multocida* disease. However, while several *P. multocida* vaccines are currently available [27], most only provide protection against strains producing the same capsule and LPS structures as the vaccine strain and the duration of immunity is generally short [28]. Currently, there is no vaccine available that protects against the full range of diseases in different hosts. We predict that to improve the breadth of protection of *P. multocida* vaccines, it is important to understand what pathogenic mechanisms are shared or unique across *P. multocida* strains that cause distinct diseases.

In this study, we investigated the pathogenic mechanisms shared between two different *P. multocida* strains, VP161, a chicken fowl cholera isolate belonging to capsule serotype A and LPS genotype L1, and M1404, a bison haemorrhagic septicaemia isolate belonging to capsule serotype B and LPS genotype L2. These strains are genetically diverse, but both can cause lethal systemic infection in mice [19,29,30]. We performed transposon-directed insertion site sequencing (TraDIS, also called Tn-Seq) in both strains to identify all genes in each strain that were essential for growth on rich media, and those genes important for *in vivo* fitness in a mouse systemic infection model. The TraDIS analysis identified a core-set of 447 genes that we predict are essential for the *P. multocida* species. We also identified shared pathways and processes important for both strains VP161 and M1404 to survive *in vivo* (35 genes), and processes important only for strain VP161 (28 genes) or only for strain M1404 (59 genes) to survive *in vivo*. These data identify multiple unique requirements for each of these strains to cause disease in mice.

## Results

### Production of a *P. multocida* strain M1404 *Himar1* mutant library, systemic infections in mice, and TraDIS library production

We previously constructed a *Himar1* transposon mutant library in the *P. multocida* strain VP161 (capsule type A and LPS type L1) [31], which was also used in this study. For TraDIS analysis of strain M1404 (capsule type B and LPS type L2), we generated a comparable *Himar1* transposon mutant library. A total of ~1.12 x 10⁶ M1404 *Himar1* transposon mutants were pooled to produce the final library. To generate a high-quality M1404 reference genome for TraDIS analysis, we used a combination of Nanopore and Illumina sequencing that allowed for the assembly of a 2.42 Mbp genome that contained two contigs (see S1 Text for full assembly methodology and S3 Text for full annotation).

The M1404 and VP161 *Himar1* mutant libraries were each used to infect BALB/c mice at a dose of ~2 x 10⁷ CFU via intraperitoneal injection (project overview described in S1 Fig, infectious dose given in S1 Table). All mice succumbed to systemic infection, reaching humane endpoints at five hours (S2A Fig). Surviving *P. multocida* mutants were recovered from the blood, liver, and spleen by plating *in vivo*-derived samples onto rich media. Greater than 4 x 10⁶ CFU of *P. multocida* were recovered from each site for all infections (S2B Fig). Four TraDIS library replicates were generated using genomic DNA isolated from the recovered mutants representing each strain (VP161 and M1404) and condition (rich media input culture, recovered mutants from each tissue following infection in mice). Each library was sequenced on an Illumina NextSeq and analysed using the Bio-TraDIS pipeline and related scripts (full TraDIS data in S1 Data). Combined, the four TraDIS library replicates generated for each strain and condition had high transposon-DNA junction read coverage across the genome and a small bp distance between unique insertion sites (UIS) (S3 Fig); the VP161 libraries had between 31 and 33 bp distance between each UIS, and the M1404 libraries had between 40 and 43 bp between each UIS (S2 Table). Principal component analysis using UIS or reads per gene showed that in all conditions the rich media TraDIS libraries clustered together and separated away from the mouse TraDIS libraries (S4 Fig).

### Identification of core genes essential for growth in rich media

We used the input mutant libraries, grown on solid heart infusion media overnight, to identify genes essential for growth in rich media for both *P. multocida* strain VP161 and M1404. Essential genes were identified by comparing normalised

UIS counts per gene (Figs 1A and S5; see S2 Data for the list of all essential genes for growth on rich media), identifying genes that resulted in lethality or a severe fitness defect when insertionally inactivated. There were 585 genes, of which 542 were coding sequences (CDS), identified as essential for strain VP161. There were also 585 genes, of which 540 were CDS, essential for strain M1404. Although a similar number of protein encoding genes were identified as essential in both strains, only 461 were homologs (Fig 1B), as determined using eggNOG or BlastP. The majority of these 461 shared essential proteins were involved in housekeeping processes. Analysis using the STRING database identified several processes and pathways that were enriched in the essential protein set, including: DNA replication (false discovery rate [FDR] = 4 x 10$^{-4}$), transcription (FDR = 0.047), translation (FDR = 2.04 x 10$^{-16}$) including ribosomal structural proteins (FDR = 2.7 x 10$^{-5}$) and tRNA amino acylation (FDR = 7.83 x 10$^{-5}$), cell division and regulation of cell shape (FDR = 0.015), fatty acid biosynthesis (FDR = 4 x 10$^{-6}$), peptidoglycan (FDR = 0.004) and lipid A biosynthesis (FDR = 0.019), glycolysis (FDR = 0.026), purine nucleoside biosynthesis (FDR = 0.0026), heme/porphyrin biosynthesis (FDR = 0.027), isoprenoid biosynthesis (FDR = 0.027), and ATP production (FDR = 0.0033).

We then assessed if these shared essential genes were also conserved across the entire *P. multocida* species. We analysed 167 publicly available complete *P. multocida* genomes, together with VP161 and M1404, and identified a total of 1,564 genes present in all strains (the 100% core-genome). A core-genome phylogeny generated from this set of 169 complete genomes was highly similar to a previously constructed *P. multocida* core-genome phylogeny generated using 402 *P. multocida* strains that included complete and incomplete genomes (S6 Fig, see S2 Text for newick format file) [7], indicating that this subset of 169 strains is representative of the entire *P. multocida* species. Of the 461 essential genes identified in both strain VP161 and strain M1404, 447 (97%) were present in the 100% core-genome (Fig 1B). We also compared the 100% core-genome and conserved rich media essential genes to our curated *P. multocida*-specific virulence factor and antibiotic resistance database (PastyVRDB) [7]. This analysis identified 53 genes associated with virulence that are present in all *P. multocida* strains, and included the genes *hpbA*, *oma87*, *spoT*, *fur*, Pm0442, and *kdtA*, which were all essential in both strains (S3 Table).

There were 75 genes essential in VP161 that had a non-essential homolog in M1404, and 65 genes essential in M1404 that had a non-essential homolog in VP161 (Fig 1B). Genes identified as essential in VP161 but not M1404 included *crp*, *crr* and *relA*, and genes identified as essential in M1404 but not VP161 included several genes involved in the Tol-Pal

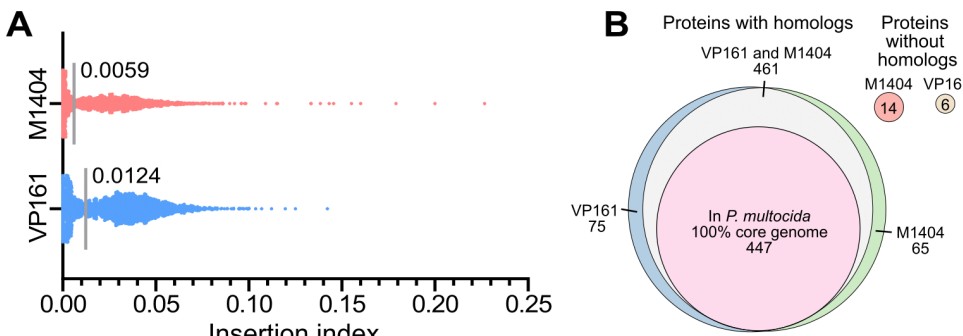

**Fig 1. Identification and comparison of genes essential for *P. multocida* strains VP161 and M1404 to grow in rich media.** Both the VP161 and M1404 *Himar1* mutant libraries were grown to mid-exponential phase in heart infusion (HI) broth, before being plated on HI agar. Viable mutants were recovered, used to generate TraDIS libraries that were sequenced by Illumina sequencing, and the data analysed using Bio-TraDIS and related scripts. **A.** Insertion indexes (number of unique *Himar1* transposon insertion sites divided by gene length) for all genes following growth in HI broth. The cut-off for essentiality is marked by a grey line for each dataset. **B.** Comparison of essential protein encoding genes present in both strain VP161 and M1404, or only one of the strains. Protein homologs between strains were identified using eggNOG matches, or by having > 90% amino acid identity. A 100% *P. multocida* core-genome was identified in strains VP161 and M1404 along with 167 complete, publicly available *P. multocida* genomes using Roary, and compared using a > 90% amino acid identity cut-off to the essential proteins identified in both strains VP161 and M1404.

system (*tolBQR* and *pal*), DNA repair (*ruvBC* and *aroQ*), and sugar transport (*ptsH*). There were also six essential VP161 genes that had no homolog in M1404, and 14 essential genes in M1404 that had no homolog in VP161 (Fig 1B). Most of the unique essential M1404 genes encoded hypothetical proteins; however, M1404_00858 and M1404_00859 contained antitoxin-like domains, and M1404_01327 contained a putative peptidase toxin domain.

### *P. multocida* strains VP161 and M1404 have both shared and unique genes required for virulence in mice

To identify genes that were important for *P. multocida* fitness during murine systemic infections, we compared the read count per gene between the mouse and rich media TraDIS libraries. Any gene with a four-fold decrease in *Himar1* insertions and a *q*-value < 0.01 was identified as resulting in decreased *in vivo* fitness when disrupted, and any gene with a four-fold increase in *Himar1* insertions and a *q*-value < 0.01 was identified as showing increased *in vivo* fitness when disrupted (see S3 and S4 Data for full fitness-associated data). Overall, 134 genes were *in vivo* fitness-associated in at least one type of tissue and one strain (Table 1 and Fig 2A). There were 63 and 94 genes that resulted in decreased *in vivo* fitness of strain VP161 and M1404, respectively (Fig 2B and S4 Table). For both VP161 and M1404, most of these genes were identified as important for *in vivo* fitness in all three tissues (Fig 2B). The genes were then compared to identify homologs between the two strains. Of the genes that had a homolog in the other strain, 35 were important for fitness in both strains, 23 were important for fitness only in VP161, and 47 were important for fitness only in M1404 (Figs 2C and 3). Furthermore, there were five genes important for fitness in VP161 that did not have a homolog in M1404, and 12 genes important for fitness in M1404 that did not have a homolog in VP161 (Figs 2C and 3). In total, there were seven genes that increased fitness when disrupted in either VP161 or M1404, with only two genes, *yeeZ* and *speB*, common to both strains (Fig 3 and S5 Table).

Common pathways and processes were identified as important for both VP161 and M1404 during infection in mice (Figs 3A, S7 and S8). In both strains, disruption of almost all capsule biosynthesis genes had a severe fitness-cost. Both *fis* and *hfq*, which encode transcriptional regulators known to positively regulate capsule biosynthesis genes in strain VP161, were important for fitness in VP161 but not in M1404, suggesting these strains used different regulatory mechanisms. Several genes encoding proteins involved in LPS core-polysaccharide synthesis were identified as important for *in vivo* fitness. Genes from the ADP-L-glycero-β-D-manno-heptose synthesis pathway (*hldDE-gmhAB*), that encode proteins that synthesise a modified heptose that is added to the LPS core-polysaccharide, were either important for *in vivo* fitness or essential in both strains. The gene *hptA*, which encodes an LPS inner-core heptosyltransferase, was identified as important for fitness in both strains. Additionally, genes encoding the LPS inner-core heptosyltransferase genes *hptC* and *hptD*, the LPS inner-core glucosyltransferase *gtcD*, and the LPS outer-core galactosyltransferase *gatD* were important for fitness of strain M1404. Several genes identified as important for *in vivo* fitness in both strains encoded proteins involved in amino acid synthesis and transport, or protein translation (Fig 3). These included an aminotransferase involved in aspartate, phenylalanine, and tyrosine synthesis (*aspC*), an asparagine synthase (*asnA*) and regulator of asparagine synthase expression (*asnC*), a putative alanine transporter (*alsT_1*), and proteins involved in translation efficiency (*prmC* and *epmA*) (Fig 3). Two genes in the L-serine biosynthesis pathway (*serAB*) and a putative tyrosine transporter (*tyrP_1*) were important for the *in vivo* fitness of VP161 but not M1404. Conversely, elongation factor Tu subunit genes (*tufB_1* and

**Table 1. Number of genes identified in *P. multocida* strain VP161 or M1404 that when disrupted by a *Himar1* transposon insertion resulted in decreased or increased *in vivo* fitness.**

|  | VP161 | | | | M1404 | | | |
|---|---|---|---|---|---|---|---|---|
|  | Blood | Liver | Spleen | Total | Blood | Liver | Spleen | Total |
| Decreased fitness | 59 | 57 | 50 | 63 | 85 | 75 | 74 | 94 |
| Increased fitness | 4 | 6 | 6 | 7 | 6 | 5 | 4 | 7 |

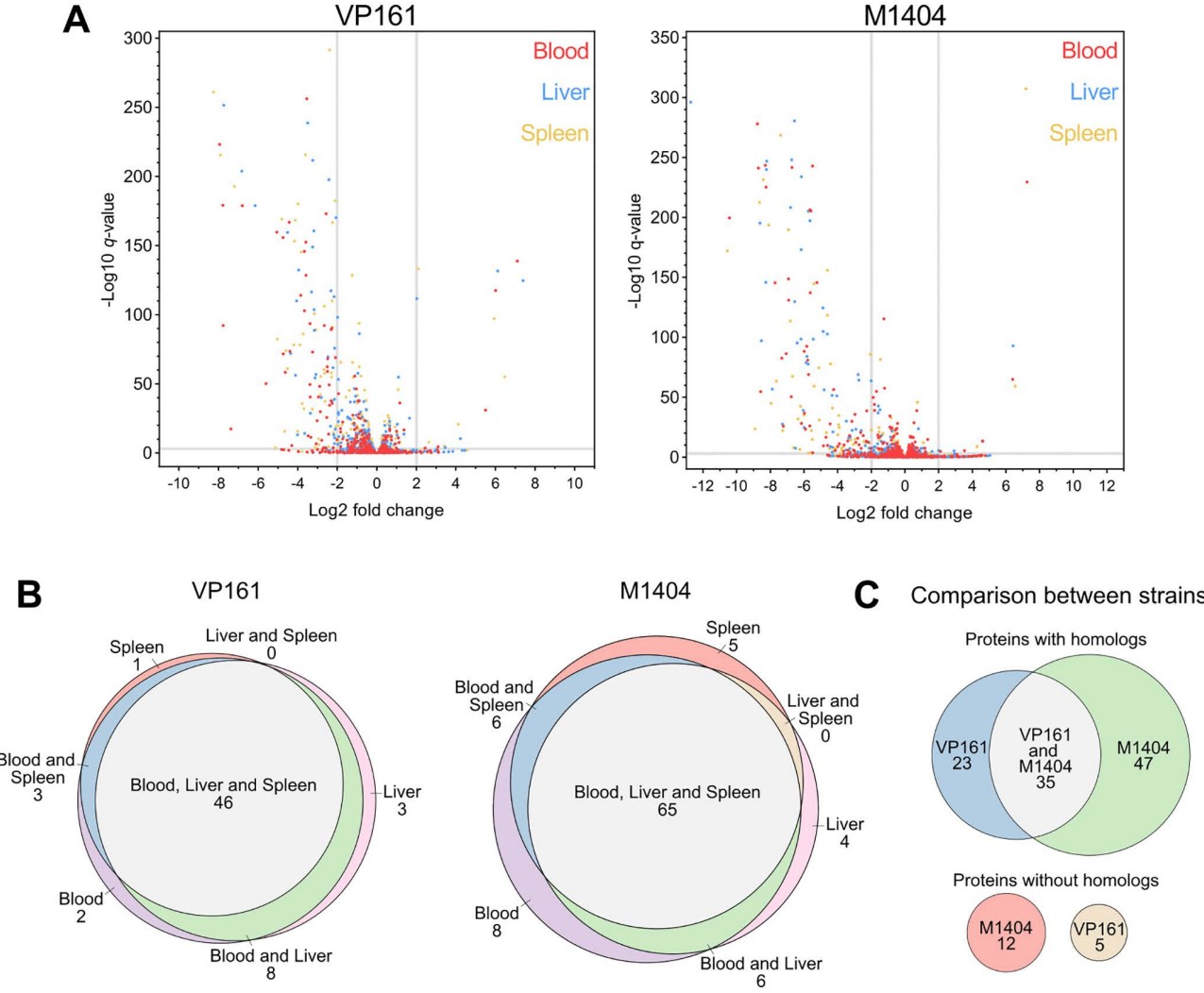

**Fig 2. Identification and comparison of genes associated with *in vivo* fitness for *P. multocida* strain VP161 and M1404 during systemic infections in mice.** The *P. multocida* strain VP161 and M1404 *Himar1* libraries were individually used in systemic infections in BALB/c mice. Surviving mutants were recovered from the bloodstream, liver and spleen, and then plated on heart infusion agar. Genomic DNA was extracted from surviving mutants and used to generate TraDIS libraries. TraDIS libraries were sequenced via Illumina sequencing and the data analysed using Bio-TraDIS and related scripts. **A.** Volcano plots showing the $\log_2$ fold-change in read count per gene and $-\log_{10}$ *q*-value for all genes in the bloodstream (red), liver (blue), and spleen (gold) TraDIS libraries compared to the rich media TraDIS library. Genes were identified as *in vivo* fitness-associated if they had a four-fold increase or decrease in read count, with a *q*-value of < 0.01 (cut-offs shown as grey lines). **B.** Venn diagrams showing the number of genes identified as important for *in vivo* fitness that were shared between the bloodstream, liver, and spleen TraDIS libraries. **C.** Comparison of *in vivo* fitness genes between strain VP161 and M1404. Protein homologs between strains were identified using matching eggNOG IDs, or by having > 90% amino acid identity and > 90% coverage.

*tufB_2*) were important for the *in vivo* fitness of strain M1404 but not strain VP161. Other systems important for *in vivo* fitness in both strains included the purine biosynthesis pathway, the Ton-B energy transduction system (*tonB-exbBD*), methionine uptake (*plpB-metPN*), and *aroA* from the chorismate pathway.

Several pathways were identified as important for *in vivo* fitness of M1404 but not VP161 (Fig 3A). This included genes encoding two components of the sialic acid uptake system (*nanP* and *nanU*), sialic acid metabolism (*nanE*) and the N-acetylneuraminate CMP transferase (*neuA_1*), a sensor histidine kinase (*cpxA*), the sorbitol-specific EIIA (*srlB*), and

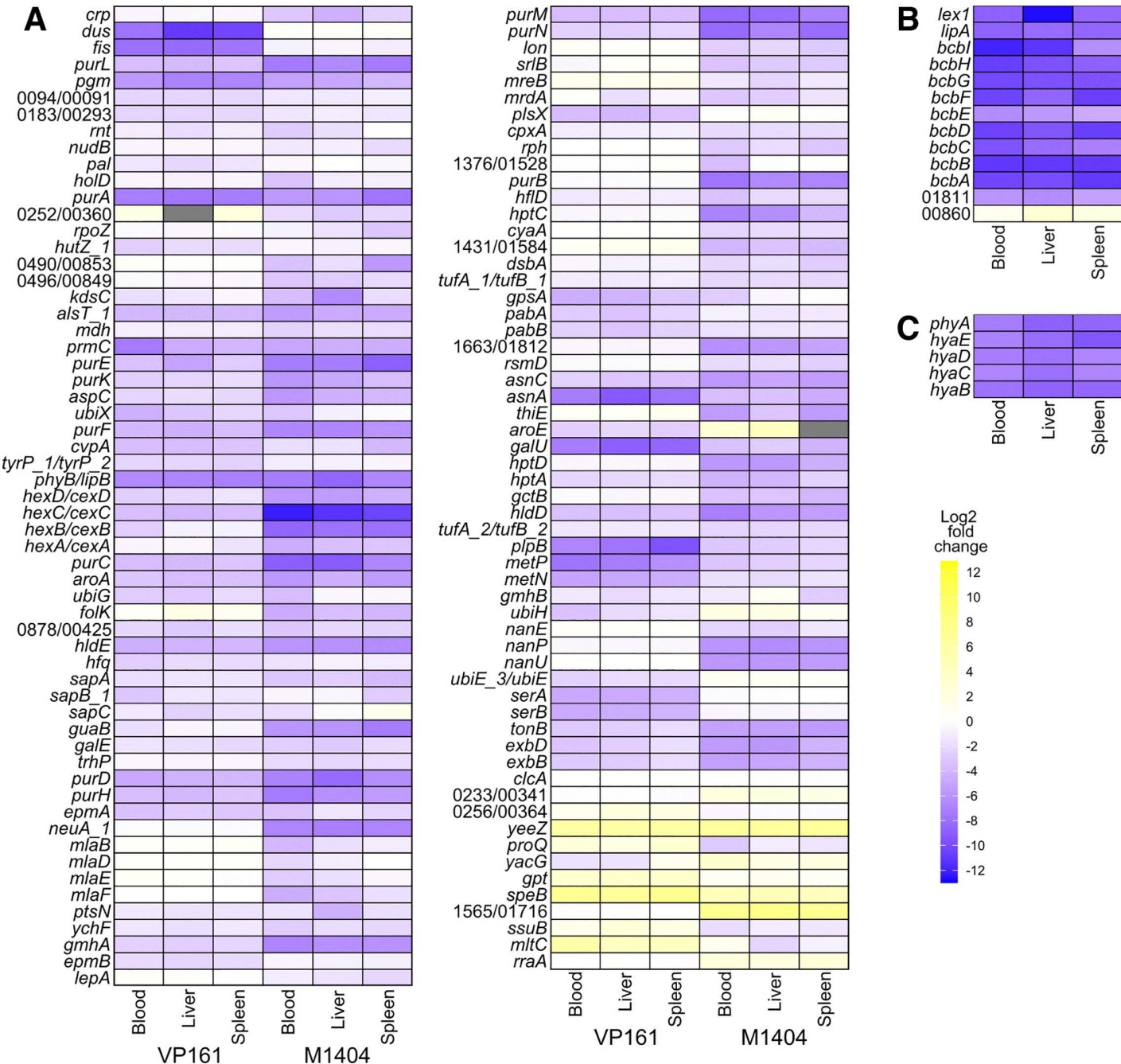

**Fig 3. Heat maps showing the log₂ fold-change in reads for *in vivo* fitness-associated genes.** The log₂ fold-change in reads per gene was identified using the fitness.comparison R-script, using the rich media TraDIS data as the control, and each of the bloodstream, liver, and spleen TraDIS libraries as the condition. Genes were included in the heatmap if they were identified as *in vivo* fitness-associated in any one of the TraDIS libraries. The gene *clcA*, which is known not to impact *in vivo* fitness [70], was included as a control. **A.** *In vivo* fitness-associated genes that are present in both strains. For genes that are annotated with different names or are identified only by different locus tags, both names or locus tags are given, with the VP161 name or locus tag first followed by the M1404 name or locus tag. For the locus tags, the prefix (PmVP161_ or M1404_) has been omitted. **B.** *In vivo* fitness-associated genes in M1404 with no homolog in VP161. **C.** *In vivo* fitness-associated genes in VP161 with no homolog in M1404.

genes encoding the Mla phospholipid transport system (*mlaBDEF*). The genes encoding adenylate cyclase (*cyaA*) and cAMP receptor protein (*crp*) were also identified as important for *in vivo* fitness of M1404 but not VP161; however, *crp* was identified as essential for strain VP161 to grow in rich media. The gene M1404_01811, which encodes a *P. multocida* capsule type B- and E-specific surface lipoprotein (SLP) PmSLP-3 [32], and the adjacent gene M1404_01812 that encodes a surface lipoprotein assembly modulator (Slam) required for PmSLP-3 export, were both identified as important for M1404 *in vivo* fitness. Comparatively, other putative SLPs (PlpE and PmVP161_1666 in strain VP161, M1404_01814 in strain M1404), and the Slam exporter in VP161 (PmVP161_1663) were not important for *in vivo* fitness. Most of the genes important for *in vivo* fitness of M1404 that did not have a homolog in VP161 were required for production of the type B capsule polymer (Fig 3B). Genes required for *in vivo* fitness of VP161 but not for M1404 included genes encoding components of the ubiquinone biosynthesis pathway (*ubiGHE_3X*), a heme utilisation protein (*hutZ_1*), components of the aminodeoxychorismate synthase complex (*pabAB*), a putative sodium proton antiporter (PmVP161_0183), a putative tyrosine importer (*tyrP_1*), and *aroE* from the chorismate pathway (Fig 3A). All of the genes important for *in vivo* fitness in VP161 that did not have a homolog in M1404 were required for production of the type A capsule polymer (Fig 3C).

### The genes *alsT_1*, *crp* and *cyaA* are important for strain M1404 fitness *in vivo*

We performed time to euthanasia experiments with individual *P. multocida* directed mutants to confirm that a sub-set of genes identified by TraDIS were individually important for *in vivo* survival. ClosTron mutagenesis was used to generate individual *alsT_1*, *crp* and *cyaA* mutants in strain M1404. Wild-type M1404 and each mutant strain was used to inoculate separate groups of BALB/c mice via intraperitoneal injection on two different days (see S1 Table for infectious doses). Signs of systemic infection were monitored and used to determine time to euthanasia. Mice infected with wild-type M1404 reached the humane endpoint on average 6.5 hours post infection (hpi) (Fig 4). Mice infected with the *crp* or *cyaA* mutants reached the humane endpoint at an average 11 and 10.3 hpi, respectively (Fig 4). Mice infected with the *alsT_1* mutants did not reach the humane endpoint and showed low total clinical scores over the 12-hour infection period (Fig 4). All strains had a significant increase in time to euthanasia when compared to wild-type M1404, agreeing with the TraDIS data that *alsT_1*, *crp* and *cyaA* are important for *in vivo* fitness of strain M1404.

### The carbon catabolite repression system controls capsule production differently in *P. multocida* strains VP161 and M1404

Carbon catabolite repression was important for *P. multocida* fitness and is controlled by the cyclic-adenosine monophosphate (cAMP) receptor protein (CRP, also called catabolite activator protein (CAP) or virulence factor regulator (VFR) in other species). CRP was important for *in vivo* fitness of strain M1404, and essential for strain VP161 to grow in rich

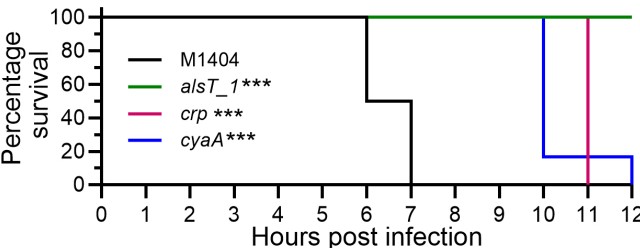

**Fig 4. Kaplan-Meier survival curves for mice infected with *P. multocida* wild-type M1404 or an M1404 directed mutant.** Groups of six BALB/c mice (three male and three female) were inoculated via intraperitoneal injection with wild-type strain M1404, the M1404 *alsT_1* mutant, the M1404 *crp* mutant, or the M1404 *cyaA* mutant (colony forming units ranged between $2 \times 10^4$ and $2 \times 10^5$). Statistical differences in survival between strains was determined using a log-rank Mantel-Cox test; *** $P<0.001$.

media. While CRP has not been well characterised in *P. multocida*, in other bacterial species CRP family proteins are global transcriptional regulators that have increased affinity for DNA targets when bound to cAMP [33]. Thus, CRP activity is modulated by the intracellular levels of cAMP. The level of intracellular cAMP is controlled by the adenylate cyclase CyaA, which synthesises cAMP from ATP [34], and the cAMP phosphodiesterase CpdA, which degrades cAMP to AMP [35]. As shown above, both *crp* and *cyaA* were identified as important for *in vivo* fitness of M1404 by TraDIS analysis and individual mutants in these genes showed reduced virulence in mice. Furthermore, the TraDIS data showed that inactivation of *cpdA* in M1404 slightly improved fitness during *in vivo* growth (average $\log_2$ fold-change of between 0.93 and 1.34, see S1 Data). Collectively, these data suggest that cAMP-dependent activation of CRP is important for M1404 *in vivo* fitness. CRP family proteins are known to regulate a wide range of targets in different bacterial species [36–38], and transcriptomic analysis of a *P. multocida* strain 0818 *crp* mutant identified reduced capsule biosynthesis gene expression [39]. To determine if inactivating the catabolite repression system affected capsule production in M1404, capsule production was assessed using cells grown overnight on solid media. Both the M1404 *crp* and M1404 *cyaA* mutants harbouring empty vector produced barely detectable absorbance readings, indicating that these strains are essentially acapsular (Fig 5A). Providing the *crp* mutant with a wild-type copy of *crp in trans* led to a modest, but significant increase in absorbance readings, compared to the *crp* mutant harbouring empty vector (Fig 5A). Providing the *cyaA* mutant with the *cyaA* complementation plasmid restored absorbance readings, and therefore capsule production, to wild-type levels (Fig 5A). These data confirm that adenylate cyclase is required for capsule production in M1404 and strongly suggest that this is a CRP-dependent mechanism.

The role of active CRP and catabolite repression in VP161 was different from that observed in M1404. While *crp* was essential for strain VP161 growth in rich media, the TraDIS data showed that inactivation of *cyaA* had no impact on *in vivo* fitness of VP161, and inactivation of *cpdA* slightly reduced fitness (average $\log_2$ fold-change of between -1.75 and

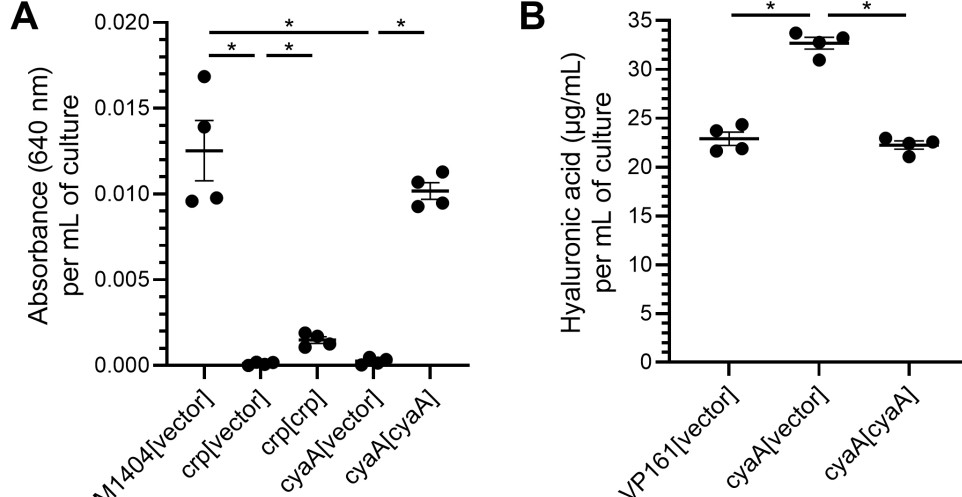

**Fig 5. Capsule produced by VP161 and M1404 wild-type and mutant strains.** Capsule absorbance assays were performed by mixing capsule extracted from cells with stains-all solution and measuring the absorbance at 640 nm. **A.** Capsule measured from wild-type M1404 harbouring empty vector, the M1404 *crp* mutant and M1404 *cyaA* mutant harbouring either empty vector or the appropriate complementation plasmid. Capsule was extracted from cells recovered from overnight growth on heart infusion agar. **B.** Capsule measured from wild-type VP161 harbouring empty vector, and the VP161 *cyaA* mutant harbouring empty vector or an expression plasmid containing an intact copy of the VP161 *cyaA* gene. Capsule was extracted from cells grown in heart infusion broth to early stationary phase (7 h), and compared to known amounts of purified hyaluronic acid. For both assays, absorbance or hyaluronic acid was standardized to value per 1 mL of culture at an $OD_{600}$ of 1. Error bars represent mean ± standard error of the mean, and statistical differences in capsule production between samples were determined using Mann-Whitney U-tests; * $P < 0.05$.

-1.93, see S1 Data). In our previous TraDISort study, *cpdA* was identified as important for *in vitro* capsule production in strain VP161 [31]. Collectively, these data suggest that active CRP-cAMP may act to repress capsule production in strain VP161. As CRP was essential in VP161, we attempted to inactivate *cpdA* and *cyaA* using ClosTron mutagenesis; however, only a *cyaA* mutant was successfully generated. Capsule absorbance assays were performed using wild-type VP161 and the *cyaA* mutant harbouring empty vector, and the complemented *cyaA* mutant. Capsule was extracted from cells during early stationary phase, when cAMP levels would be expected to be high. The VP161 *cyaA* mutant produced significantly more capsule than wild-type VP161, with complementation restoring normal levels of capsule production (Fig 5B). These data indicate that inactivation of adenylate cyclase, which should reduce the level of cAMP and therefore CRP activity, de-represses capsule production in VP161. This strongly suggests that an active CRP-cAMP represses capsule in strain VP161, which is the opposite of the role of CRP-cAMP in M1404.

## Discussion

In this study, we performed TraDIS analysis of two distinct *P. multocida* strains, the poultry isolate VP161 (A:L1) and the bovine isolate M1404 (B:L2). These strains produce different capsule and LPS structures. Initially, we identified genes required for both VP161 and M1404 to grow in rich media, with 585 genes identified as essential in both strains. We have previously reported that 509 genes were essential for VP161 to grow in rich media [31], 95% of which were also identified as essential in this study. The increase in rich media essential genes is likely due to the mutant libraries being grown for ~48 h longer than the previous study, allowing for further reduction of mutants with reduced viability from the population. Previous analysis of random transposon mutant viability has shown that the time required for a mutation to lead to a replication defect determines at what time point the gene is identified as important for survival [40]. Strains VP161 and M1404 shared a set of 461 protein-encoding genes that were essential for growth in rich media, most of which were involved in housekeeping processes. Pan-genome analysis of 169 complete *P. multocida* genomes identified 1,564 genes present in all genomes. Of these 1,564 core genes, 447 were identified as essential in both strain VP161 and strain M1404 for growth in rich media. Thus, 97% of the 461 essential genes identified in strains VP161 and M1404 are present in all *P. multocida* genomes examined and are likely to be required for all *P. multocida* strains to grow in rich media. While most of these genes encode housekeeping genes involved in replication, transcription, translation and energy metabolism, some encode surface-associated proteins and therefore could be considered potential targets for vaccine development.

TraDIS analysis of VP161 and M1404 mutants recovered from BALB/c mice following systemic infection identified 134 genes associated with *in vivo* fitness in at least one tissue and one strain. To validate these results, we generated directed *alsT_1*, *crp* and *cyaA* mutants in strain M1404. Separate infection studies with these mutants showed reduced virulence in mice, suggesting each is important for *in vivo* fitness. Complementation was not performed to reduce the number of mice used in this study. Additionally, several genes and pathways identified in the TraDIS analysis performed in this study have previously been identified as important for *P. multocida* virulence. Infection studies conducted using mice or poultry have shown that inactivation of *aroA* in *P. multocida* strains X-73, P-1059, and 85020 [41,42], *tonB*, *exbB*, and *exbD* in strain PM25 [43], *plpB* in strain VP161 [44], and *nanP* and *nanU* in strain P-1059 [45] all resulted in reduced virulence. Furthermore, several *P. multocida aroA* mutants are commercially available for use as live-attenuated vaccines [41,42]. Signature-tagged mutagenesis studies using strain UC6731 and strain VP161 identified *purN*, *purF*, *guaB*, *nanE* and *nanU* as important for virulence in mice and/or chickens [29,46]. Transcriptomics analysis of strain X-73 cells recovered from systemic infections in chickens showed there was increased expression *in vivo* of *purH*, *aspC* and *asnA* [47], and transcriptomic analysis of Pm70 grown in media with limited iron showed increased expression of *hutZ_1* [48]. These previous studies strongly support that our TraDIS analysis has comprehensively identified genes required for *P. multocida* survival *in vivo*. Several biological processes were identified as important for *P. multocida* survival that have also been identified by TraDIS as important for pathogenesis or *in vivo* survival in other bacterial species, including methionine uptake and purine biosynthesis in *Yersinia pestis* [49], and the shikimate pathway in *Salmonella enterica* subsp. Typhimurium [50]. Several

surface structures that have previously been proposed to play a role in *P. multocida in vivo* fitness and/or virulence, including filamentous haemagglutinin, fimbriae, and Flp-pili, were not identified as important for *in vivo* fitness in our TraDIS analysis [29,46,51,52]. These surface structures are likely required for initial colonisation of the upper respiratory tract but not when *P. multocida* is delivered via intramuscular, intraperitoneal or intravenous injection. Indeed, *P. multocida* filamentous haemagglutinin mutants have a higher infectious dose (ID$_{50}$) compared to wild-type parent strains in birds but only when introduced intranasally and not via intramuscular or intravenous route [51,52]. The TraDIS data presented in this study identified several processes that have not previously been shown to be important for *P. multocida* disease, including ubiquinone biosynthesis, aminodeoxychorismate synthesis, the Mla phospholipid transport system and putative sodium proton antiporters. Importantly, we showed that there were clear differences between pathways and processes required for the two different *P. multocida* strains to survive *in vivo*. The ubiquinone biosynthesis pathway and the putative sodium proton antiporter (PmVP161_0183) were only important for *in vivo* fitness of strain VP161. Conversely, components of the sialic uptake and utilisation pathway, the Mla ABC-transport system, the adenylate cyclase CyaA, the SLP protein PmSLP-3 and its cognate Slam exporter were only important for *in vivo* fitness of strain M1404. This crucially shows that although these *P. multocida* strains share these systems, their importance for survival *in vivo* is strain dependent.

Different *P. multocida* strains typically encode two or more SLPs that are exported to the cell surface by a Slam protein [32,53]. Subunit vaccines formulated using *P. multocida* SLPs (PlpE, PmSLP-3 and PmSLP-1) have been shown to elicit a protective immune response against *P. multocida* strains that produce the cognate SLP [32,54–56]. Despite both VP161 and M1404 encoding two SLPs, our TraDIS analysis only identified PmSLP-3 in M1404 as important for survival in BALB/c mice. PmSLP-1, an SLP that is encoded by a subclade of capsule type A strains isolated from cattle, has recently been shown to bind to complement factor I and block complement-mediated killing [57]. Previous studies have shown that capsule was important for blocking complement-mediated killing in strain VP161 but not for strain M1404 [19,30]. As such, it is possible that PmSLP-3 is important for blocking complement-mediated killing in M1404. Several SLPs from *Neisseria*, *Actinobacillus*, *Hemophilus* and *Moraxella* spp. have a similar predicted protein fold to *P. multocida* SLPs but have diverse functions, including binding to human iron-binding proteins, lactoferrin and factor H [53]. Thus, despite SLPs having similar structure, each protein can have a different biological function and role in pathogenesis. Given *P. multocida* can persist in several different hosts, it is possible that different SLAM-dependent SLPs are required for survival in different hosts.

Almost all capsule biosynthesis genes and known regulators of capsule biosynthesis were identified as important for *in vivo* fitness, supporting many previous studies demonstrating that capsule is important for *P. multocida* virulence in both type A and B strains [19,30,46,58–60]. Additionally, several other genes previously identified in strain VP161 as important for capsule production were also identified by TraDIS as important for *in vivo* fitness, including *prmC*, *epmA*, *epmB* and *ubiX*. However, our data indicated that capsule production is regulated differently in the two strains. The genes encoding the global regulator Fis and the RNA chaperone protein Hfq, previously shown to be important for capsule production in strain VP161 [59,60], were also identified in this study as important for *in vivo* fitness of VP161 but not M1404. Furthermore, the TraDIS analysis and capsule absorbance assays together showed that CRP-cAMP controls capsule production in these *P. multocida* strains, but in different ways. Disrupting CRP-cAMP by inactivating either *cyaA* or *crp*, abrogated capsule production in strain M1404, showing that CRP-cAMP positively regulates capsule production in this strain. Conversely, disrupting CRP-cAMP by inactivating *cyaA* increased capsule production in strain VP161, indicating that CRP-cAMP represses capsule production in strain VP161. Together, these data show that capsule expression responds differently to CRP-cAMP in these two strains. CRP/VFR has been shown to regulate different virulence factors in other bacterial species [36,61,62], and has been identified as important for *K. pneumoniae* fitness during bacteraemia by TraDIS [63]. The TraDIS data also suggested that activation of the catabolite response in strain M1404 may depend on the level of sorbitol. Adenylate cyclase activity is commonly controlled by phosphoenolpyruvate(PEP):carbohydrate phosphotransferase system (PTS) proteins. The PTS system couples carbohydrate import and phosphorylation, with the energy required for import and the donated phosphate provided by PEP [64,65]. The phosphate is transferred to the carbohydrate

via a protein phosphate transfer chain consisting of the general PTS proteins EI and HPr, and the sugar specific EII, with EII generally comprised of three proteins/domains (EIIA, EIIB, EIIC) [66]. Different proteins in the PTS system can allosterically regulate different protein targets depending on whether they are phosphorylated. In *E. coli*, phosphorylated EIIA$^{Glc}$ is known to be involved in inducing adenylate cyclase activity [67,68], thus allowing cAMP production and activation of the catabolite repressor system when the levels of glucose are low. EIIA$^{Glc}$, encoded by *crr*, was not important for *in vivo* fitness of M1404. However, the sorbitol-specific EIIA protein, encoded by *srlB*, was identified as important for *in vivo* fitness of M1404. This suggests that the absence of sorbitol increases the amount of phosphorylated sorbitol-specific EIIA, which in turn induces adenylate cyclase activity and activates the catabolite response.

Several LPS biosynthesis genes were important for fitness in both strains. The LPS of *P. multocida* lacks an O-antigen, making the core-polysaccharide the most distal component of the LPS [69]. The core-polysaccharide is comprised of a conserved inner-core and a variable outer core, and is synthesised by stepwise addition of sugars by different glycosyltransferases [69]. As the function of LPS transferases have been determined for LPS type L1 and L2 strains [70–73], the putative structures required for survival in BALB/c mice can be predicted based on which transferases had reduced fitness in the TraDIS analysis. All *P. multocida* strains examined to date simultaneously produce two inner-core structures (namely glycoforms A and B), with the relative levels of each glycoform determined by the competing activity of KdkA and KdtA. KdkA adds a phosphate to the first 3-deoxy-D-manno-oct-2-ulosonic acid (Kdo) leading to glycoform A production, while KdtA adds a Kdo II to Kdo I, leading to glycoform B production [70]. The nascent inner-core is then extended by the heptosyltransferases HptA (specific for glycoform A) and HptB (specific for glycoform B) [70]. The TraDIS analysis identified *hptA*, but not *hptB*, as important for fitness in both strain VP161 and M1404. Inactivation of *hptA*, but not *hptB*, has previously been shown to attenuate virulence of strain VP161 in chickens [70]. The VP161 *hptA* mutants have severely truncated LPS as the Lipid A – phosphorylated Kdo I cannot be extended in this mutant [70]. The presence of a large amount of severely truncated LPS on the surface make the bacterium vulnerable to the innate immune system [70]. No other LPS transferases were identified as important for fitness of VP161, supporting previous data that showed a full length LPS is not important for fitness of this strain in mice [29]. In strain M1404, the inner-core transferases *hptC*, *hptD* and *gctA* were all important for fitness, indicating that a glycoform A structure of at least Kdo I - Hep I [Hep II – Hep III] – Glucose (Glc) I is required for *in vivo* survival. Identification of *gmhB*, involved in ADP-heptose synthesis, was identified as important for *in vivo* fitness of both strains. Previous studies have shown that *gmhB* is important for fitness of several Enterobacterales species in mice [74,75], indicating that ADP-heptose synthesis is likely important for virulence in several Gammaproteobacteria species. Additionally, inactivation of the outer-core galactosyltransferase gene *gatD* significantly reduced fitness of M1404 in BALB/c mice. The LPS outer-core structure produced by M1404 is Hep IV – Hep V – N-acetyl glucosamine (GlcNAc) I – Galactose (Gal) I [73]. Identification of *gatD* but not other outer-core transferases as important for fitness suggests that removal of the terminal galactose, therefore leaving the GlcNAc as the terminal sugar, reduces the viability of M1404 in BALB/c mice, but truncations terminating the outer-core at either Hep IV, Hep V or GlcNAc I do not impact fitness. Previous analysis of several *P. multocida* type L3 strains has shown that inactivating mutations in transferase genes that result in different LPS outer-core structures affect virulence in chickens at different levels [76].

In this study, we have determined the 100% core-genome for *P. multocida* and identified a set of core genes likely essential for all *P. multocida* strains. Furthermore, we have comprehensively identified genes important for *in vivo* fitness of the type A strain VP161 and the type B strain M1404 during systemic infections in BALB/c mice, with these strains having shared and unique pathways that are important for fitness. Genes identified as important for both strains to survive *in vivo* and conserved across the entire *P. multocida* species represent ideal vaccine targets for generating a vaccine that provides protection against all *P. multocida* strains. Finally, we have shown that capsule production is regulated differently between these two strains, and that a full length LPS is more important in M1404 than VP161 for *in vivo* fitness in mice.

## Materials and methods

### Ethics statement

Animal experiments were performed according to Victorian State Government regulations, approved by the Monash University Animal Ethics Committee (Monash University AEC no. MARP/2023/36738).

### Bacterial strains and growth conditions

The bacterial strains and plasmids used in this study are listed in S6 Table. *E. coli* strains were cultured in lysogeny broth (LB), and *P. multocida* strains were cultured in heart infusion (HI) broth (Oxoid). Solid media was generated by adding 1.5% agar (w/v) to liquid media. When required, antibiotics were added to the media at the following concentrations: kanamycin 50 µg/mL, spectinomycin 50 µg/mL, tetracycline 1.25 µg/mL, ampicillin 100 µg/mL. When required, diaminopimelic acid (DAP) was added to media at a concentration of 100 µM.

### Oligonucleotides, DNA extraction, DNA manipulation and DNA sequencing

All oligonucleotides used in this study were synthesised by Sigma, and the sequences of all oligonucleotides are given in S7 Table. Genomic DNA was extracted using the HiYield Genomic DNA Mini Kit (Real Biotech Corporation) as per the manufacturer's instructions for extraction from Gram-negative bacteria or extracted using Genomic Tip 100/G (QIAGEN) as per manufacturer's instructions. Plasmids were extracted using the NucleoSpin Plasmid Mini Kit (Machery-Nagel) as per the manufacturer's instructions. Polymerase chain reactions (PCR) were performed using either *Taq* DNA polymerase (Roche) or Phusion High-Fidelity DNA polymerase (New England Biolabs), as per the manufacturer's instructions for each polymerase. DNA digests were performed using restriction enzymes sourced from New England Biolabs as per manufacturer's instructions. Ligations were performed using T4 DNA ligase (Roche) as per manufacturer's instructions. DNA was purified using the NucleoSpin Gel and PCR Clean-up Mini Kit (Machery-Nagel). Sanger sequencing reactions were performed as previously described [76]. For whole-genome sequencing of *P. multocida* strain M1404, genomic DNA was submitted to The Ramaciotti Centre for Genomics (University of New South Wales) and sequenced using both an Illumina MiSeq v3, and a Nanopore GridION X5 Mk1 using a GridION flow cell (FLO-MIN112) with DNA prepared for sequencing using a Native Barcoding Expansion Kit (SQK-NBD112.24). Additionally, M1404 genomic DNA was also submitted to Plasmidsaurus and sequenced using a Nanopore PromethION using a PromethION flow cell (FLO-PRO114M) with DNA prepared for sequencing using a Rapid Barcoding Kit (SQK-RBK114.96). The M1404 genome was assembled as previously described [7]; see S1 Text for additional information.

### Transformation, site-directed mutagenesis using ClosTron and construction of complementation plasmids

Electrocompetent *P. multocida* cells were prepared and transformed as previously described [30]. *P. multocida* site-directed mutants were generated by a modified version of the ClosTron system (also called TargeTron) in the pAL953 plasmid as previously described [76,77] (see S1 Text for more detail on pAL953 modification). Intron insertions into each of the genes of interest were confirmed by PCR-amplification using primers that flanked the gene of interest, and by Sanger sequencing directly from genomic DNA extracted from putative mutants with oligonucleotide BAP6544, which binds within the intron and directs sequencing into the adjacent region of the genome. For complementation, the wild-type gene was amplified using Phusion polymerase and cloned into pAL99S under the control of the *tpiA* promoter. The sequence of the cloned gene and promoter were confirmed by Sanger sequencing. Complementation plasmids or pAL99S empty vector were used to transform the corresponding mutant strains to produce the complemented mutant and empty vector strains, respectively.

## Conjugation conditions and generation of the *P. multocida* strain M1404 *Himar1* transposon mutant library

Conjugation between an *E. coli* donor and *P. multocida* recipient was performed by filter mating as previously described [31], with minor modifications. The donor used in this study was *E. coli* strain JKE201 [78], a diaminopimelic acid (DAP) auxotrophic strain that cannot grow without the addition of DAP in the media. Filters containing the conjugation mix were incubated on HI agar plates containing DAP for one hour. The conjugation mix was recovered from the filter by vortexing the filter in 1 mL of HI broth. The conjugation mix was plated onto LB agar with ampicillin and DAP to recover donor *E. coli*, HI agar to recover all *P. multocida* recipients, and HI agar with spectinomycin to recover *P. multocida Himar1* mutants. *P. multocida Himar1* transconjugants were recovered from initial selection plates after overnight growth and replated on fresh HI agar containing spectinomycin to ensure selection against mutants with insertions in essential genomic regions, and to increase culture density for library storage. Following the second overnight growth, the *P. multocida Himar1* mutants were recovered in glycerol broth (30% glycerol in HI broth) and stored at -80°C. A total of ~1.12 x $10^6$ M1404 *Himar1* mutants were pooled to generate the final M1404 mutant library.

## Mouse systemic infections

Each *P. multocida* systemic infection was performed with an equal number of male and female 6–10-week-old BALB/c mice (housed separately) supplied by Monash Animal Research Platform (Monash University), with mice housed in the Animal Research Laboratory (Monash University). *P. multocida* strains used for systemic infections were grown in HI medium to early-exponential phase ($OD_{600}$ ~0.25). Mice were each injected intraperitoneally using a 27-gauge needle with a 200 μL total volume containing one of the following: ~1 x $10^7$ CFU of the *P. multocida* VP161 transposon mutant library, ~1 x $10^7$ CFU of the *P. multocida* strain M1404 transposon mutant library, ~5 x $10^4$ CFU of wild-type strain M1404, or ~1 x $10^5$ CFU of one of the directed M1404 mutants (see S1 Table). Mice were monitored using a clinical score monitoring system for systemic infections in mice [79]; mice were monitored hourly and reached humane endpoints when they had a clinical score of 4 in a single category, or a cumulative clinical score of 10 or more. Mice were humanely euthanised using $CO_2$ asphyxiation followed by cervical dislocation. Blood samples were collected via cardiac puncture, and liver and spleen samples were recovered and homogenized. Tissue and blood samples were plated at the appropriate dilutions onto HI agar to obtain single colonies and determine the bacterial burden in the bloodstream or tissue. For TraDIS analysis, to ensure there were enough cells for genomic DNA extraction, blood and homogenized tissue samples were plated neat onto heart infusion agar and grown overnight before bacteria were recovered. Samples of the VP161 and M1404 *Himar1* mutant libraries used to inoculate the mice were also plated neat onto HI agar in the same way as the bacteria recovered from mice. For DNA isolation, bacterial samples recovered from solid media using 1 x PBS were centrifuged at 13,000 x *g* then the cell pellet was resuspended 1 x PBS to wash cells before being centrifuged again as above. The cell pellets generated were then used to extract genomic DNA.

## Transposon-directed insertion site sequencing library production, Illumina sequencing and bioinformatic analysis of TraDIS libraries

TraDIS libraries were prepared for Illumina sequencing as previously described [31]. TraDIS libraries were sequenced by The Ramaciotti Centre for Genomics at the University of New South Wales using an Illumina NextSeq1000 P1 with 100 bp-single end reads. TraDIS data was analysed using the Bio-TraDIS toolkit and modified scripts, and replicates compared by principal component analysis, as previously described [31]. Briefly, genes essential for growth in rich media were identified by comparing normalized UIS per gene excluding insertions in the final 10% of the gene. Insertion indexes for all genes were plotted on a histogram that generates a bimodal distribution. Two normal curves are drawn from this histogram, and the intersect between the two normal curves is taken to be the cut-off to determine essentiality. Genes important for fitness were identified by comparison of read count per gene between each mouse TraDIS library and the input rich medium TraDIS library. Genes with $\log_2$ fold-change<-2 and a *q*-value<0.01 were called as having a fitness cost, and

genes with a $\log_2$ fold-change > 2 and a *q*-value < 0.01 as having a fitness benefit. Full data sheets are provided as S1–S4 Data.

To identify protein homologs between strains, all coding sequences from *P. multocida* strains VP161 and M1404 were analysed using eggNOG mapper 5.0 [80] and then matched by seed ortholog value; proteins with the same seed ortholog value were taken as homologs. Proteins without a match in eggNOG and non-coding sequences were compared using the Basic Local Alignment Search Tool (BLAST) [81], with homologs called when there was > 90% amino acid/nucleotide identity over >90% of the query. Venn diagrams were generated using https://eulerr.co/. Heatmaps were generated in R using the ggplot2 package [82]. String was used to identify genes involved in the same pathway or processes, and to identify pathways and processes enriched in each dataset [83]. Roary analysis was performed using the VP161 and M1404 genomes, and an additional 167 complete *P. multocida* genomes obtained from the BV-BRC database (in April 2024). Pan-genome analysis and maximum-likelihood core-genome phylogenies were generated using Roary and IQ-TREE as previously described [7].

### Capsule absorbance assays

For capsule extractions from VP161 and derivative strains, cells were cultured in liquid broth and samples taken to assess capsule production at early stationary phase (7 h, $OD_{600}$ ~ 2.2). For capsule extractions from M1404 and derivative strains, cells were cultured overnight on heart infusion agar and then recovered by adding 1 x PBS to the plate and scraping ($OD_{600}$ ~ 15). As purified polysaccharide equivalent to the *P. multocida* type B capsule is not commercially available to use as a standard, the absorbance values were normalised against the optical density values of the culture used for each capsule extraction. Capsule absorbance assays were performed as previously described [7]. Briefly, 10 µL of extracted capsule was mixed with 90 µL stains-all solution, added to a 96-well plate and the optical density at 640 nm measured using a Tecan Infinite 200 Pro. A standard curve was generated using purified hyaluronic acid (Sigma catalogue number H5388), allowing hyaluronic acid amounts recovered from VP161 strains to be determined. For both VP161 and M1404, hyaluronic acid or absorbance values were normalised by the optical density of the input culture used to give hyaluronic acid or absorbance per 1 mL of culture at an $OD_{600}$ of 1. Statistical comparisons between strains were performed using a Mann-Whitney U-test.

### Supporting information

**S1 Text. Additional materials and methods.**
(DOCX)

**S2 Text. Newick format maximum-likelihood core-genome phylogeny of 169 *P. multocida* strains.**
(DOCX)

**S3 Text. Annotated assembly of *P. multocida* strain M1404 used for TraDIS analysis in this study.**
(DOCX)

**S1 Data. All TraDIS data from this study.**
(XLSX)

**S2 Data. TraDIS data for genes identified as essential for *P. multocida* strains VP161 and M1404 growth in rich media.**
(XLSX)

**S3 Data. TraDIS data for genes identified as having an *in vivo* fitness-cost.**
(XLSX)

**S4 Data. TraDIS data for genes identified as having an *in vivo* fitness-benefit.**
(XLSX)

**S1 Table. Infectious dose and number of mice used for *P. multocida* systemic infections performed in this study.**
(DOCX)

**S2 Table. TraDIS mapping statistics for the input transposon mutant libraries.**
(DOCX)

**S3 Table. Genes in the 100% *P. multocida* core genome (1,564 genes) that encode proteins with matches to the curated database of *P. multocida* virulence factor and antibiotic resistance genes at >75% amino acid identity.**
(DOCX)

**S4 Table. Genes that result in decreased *in vivo* fitness when disrupted in *P. multocida* strain VP161 or M1404, as identified by TraDIS analysis of mutants recovered from either the blood, liver or spleen following systemic infections in BALB/c mice.**
(DOCX)

**S5 Table. Genes that result in increased *in vivo* fitness when disrupted in *P. multocida* strain VP161 or M1404, as identified by TraDIS analysis of mutants recovered from either the blood, liver or spleen following systemic infections in BALB/c mice.**
(DOCX)

**S6 Table. Strains and plasmids used in this study.**
(DOCX)

**S7 Table. Oligonucleotides used in this study.**
(DOCX)

**S8 Table. Prophage regions identified in the *de novo* assembled M1404 genome.**
(DOCX)

**S1 Fig. Overview of the TraDIS methodology from this study.** Created in BioRender. Boyce, J. (2025) https://BioRender.com/iqy3pkk.
(DOCX)

**S2 Fig. Data from mouse systemic infections performed using the *P. multocida* strain VP161 and M1404 *Himar1* mutant libraries.**
(DOCX)

**S3 Fig. Log$_2$ read count (blue) and unique insertion sites (red) per 1 kb across the *P. multocida* VP161 and M1404 genome.**
(DOCX)

**S4 Fig. Principal component analysis (PCA) plots for all VP161 and M1404 TraDIS libraries in this study.**
(DOCX)

**S5 Fig. Histograms of unique insertion sites (UIS) per gene and identification of essential gene cut-offs in the *P. multocida* strain VP161 and M1404 rich media TraDIS libraries.**
(DOCX)

**S6 Fig. Maximum-likelihood core-genome phylogeny of the 169 complete *P. multocida* genomes used to identify the 100% *P. multocida* core genome.**
(DOCX)

**S7 Fig. STRING interaction map for *P. multocida* strain VP161 *in vivo* fitness genes.**
(DOCX)

**S8 Fig. STRING interaction map for *P. multocida* strain M1404 *in vivo* fitness genes.**
(DOCX)

**S9 Fig. Mauve alignment showing colinear parts of the 2.35 Mbp and 66 kb contigs generated in the *P. multocida* strain M1404 assembly with strains VP161 and X-73.**
(DOCX)

## Acknowledgments

We would like to acknowledge the Monash Bioinformatics Platform for assistance with bioinformatic analysis. We would like to acknowledge Abbey Allgood and Irene Alevizos for assistance provided while plating bacteria during the mouse infections.

## Author contributions

**Conceptualization:** Thomas R. Smallman, Marina Harper, John D. Boyce.

**Data curation:** Thomas R. Smallman.

**Formal analysis:** Thomas R. Smallman, Xiaochu Wang.

**Funding acquisition:** Marina Harper, John D. Boyce.

**Investigation:** Thomas R. Smallman, Xiaochu Wang, Xenia Kostoulias, Amy Wright, Faye C. Morris.

**Methodology:** Thomas R. Smallman.

**Project administration:** Thomas R. Smallman, Marina Harper, John D. Boyce.

**Supervision:** Marina Harper, John D. Boyce.

**Validation:** Thomas R. Smallman.

**Visualization:** Thomas R. Smallman.

**Writing – original draft:** Thomas R. Smallman.

**Writing – review & editing:** Thomas R. Smallman, Xiaochu Wang, Xenia Kostoulias, Amy Wright, Faye C. Morris, Marina Harper, John D. Boyce.

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
