## [Decision Letter · Decision Letter 0]

28 Aug 2025

Comparative analysis of shared and unique mechanisms important for diverse strains of Pasteurella multocida to cause systemic infection in mice

PLOS Pathogens

Dear Dr. Boyce,

Thank you for submitting your manuscript to PLOS Pathogens. After careful consideration, we feel that it has merit but does not fully meet PLOS Pathogens's publication criteria as it currently stands. Therefore, we invite you to submit a revised version of the manuscript that addresses the points raised during the review process.

Please submit your revised manuscript within 60 days Oct 27 2025 11:59PM. If you will need more time than this to complete your revisions, please reply to this message or contact the journal office at plospathogens@plos.org. Please include the following items when submitting your revised manuscript:

We look forward to receiving your revised manuscript.

Kind regards,

Sebastian Suerbaum, MD

Academic Editor

PLOS Pathogens

David Skurnik

Section Editor

Editor-in-Chief

PLOS Pathogens

Editor-in-Chief

PLOS Pathogens

orcid.org/0000-0002-7699-2064

**Additional Editor Comments:**

Your manuscript has been reviewed by three experts. All three reviewers were overall positive about your paper, but they also identified issues that must be addressed. I think the point raised by reviewer 2 about the importance of complementation is well taken. The paper could be strengthened by adding more complementation data. While it may be unrealistic to provide coomplementation data for all mutants with a phenotype in a global study such as yours, it seems important to be very transparent about statements that are backed up by additional data such as complementation and more cautious conclusions that only rely on the phenotypic analysis of the mutants without complementation. Overall, I think that the reviewers make reasonable demands that could be addressed with reasonable effort.

**Journal Requirements:**

1) Please upload all main figures as separate Figure files in .tif or .eps format. For more information about how to convert and format your figure files please see our guidelines: 

2) Some material included in your submission may be copyrighted. According to PLOSu2019s copyright policy, authors who use figures or other material (e.g., graphics, clipart, maps) from another author or copyright holder must demonstrate or obtain permission to publish this material under the Creative Commons Attribution 4.0 International (CC BY 4.0) License used by PLOS journals. Please closely review the details of PLOSu2019s copyright requirements here: PLOS Licenses and Copyright. If you need to request permissions from a copyright holder, you may use PLOS's Copyright Content Permission form.

Potential Copyright Issues:

- Figure S1. Please confirm whether you drew the images / clip-art within the figure panels by hand. If you did not draw the images, please provide (a) a link to the source of the images or icons and their license / terms of use; or (b) written permission from the copyright holder to publish the images or icons under our CC BY 4.0 license. Alternatively, you may replace the images with open source alternatives. See these open source resources you may use to replace images / clip-art:

3) Please ensure that the funders and grant numbers match between the Financial Disclosure field and the Funding Information tab in your submission form. Note that the funders must be provided in the same order in both places as well.

**Reviewers' Comments:**

Reviewer's Responses to Questions

**Part I - Summary**

Reviewer #1: This is a highly interesting manuscript that describes the application of innovative TraDIS methodology to define essential genes and genes required for systemic infection in P. multocida. Importantly, several genes identified for systemic infection were validated by the construction of mutants and subsequent infection. I particularly liked how the TraDIS methodology was applied to identify genes in the mouse model, as this is an experimental approach that has proven challenging for other pathogens. The comparative analysis of 2 different P. multocida strains is another strength of the study.

Overall, the data presented are sound, the manuscript is very well written, and work will be of enormous interest to other researchers in the field. I congratulate the authors on an excellent study.

Reviewer #2: In this manuscript by Smallman and colleagues, the authors seek to identify Pasteurella multocida genes that are requires for systemic infection in mice across two distinct species. Combining in vitro and in vivo approaches, the authors leverage TnSeq to define genes used by P. multocida to cause infection. Additionally, the authors demonstrate that two distinct strains have conserved and unique factors to cause infection. Combined, this study is an interesting survey of how an important Gram-negative species causes a relevant infection. The experiments are well performed but lack proper complementation. The study would also be more impactful if placed in the context of a broader Gram-negative systemic infection field in addition to the Pasteurella field.

Reviewer #3: In this study, the authors employed transposon-directed insertion site sequencing (TraDIS) to generate a comprehensive mutant library of strain M1404, achieving genome-wide coverage and random insertional mutagenesis. By integrating advanced bioinformatics tools such as eggNOG mapper 5.0, BLAST, and STRING, the study systematically identified essential genes in both VP161 and M1404 strains under in vitro and in vivo conditions. Furthermore, the authors conducted a comparative analysis of the shared and strain-specific essential genes, as well as multiple pathways implicated in bacterial adaptability. The research presents a novel and significant direction, utilizes a suite of state-of-the-art analytical techniques, and provides a rich and substantial dataset. The manuscript is well-structured, with a logically sound experimental design, meticulous data processing, and a clear analytical framework, reflecting a rigorous scientific approach and outstanding professional competence. Based on the content of this article, I have raised several questions and would appreciate further elaboration from the authors.

**Part II – Major Issues: Key Experiments Required for Acceptance**

Reviewer #1: I have no major issues.

Reviewer #2: Major Concerns:

1. Genetic complementation throughout the study is not sufficient. In Figure 4, the experimental design should include complementation to ensure that the shift in virulence is specifically due to the described gene and not an operon off-target effect. In Figure 5A, the crp complementation is marginal. Is this within the reliable reading range of the plate reader? There appears to only be a ~0.002 difference in absorbance which is considered well beyond the linear detectable range for most absorbance-based approaches.

2. It is exciting that many hits in this TnSeq study have been identified in other Gram-negative species. This indicates larger trends in the bacterial pathogenesis field that specific mechanisms may be conserved across species in systemic infection. In the Discussion, the authors should place their findings in the context of those from other groups who have used similar approaches in different species. A few examples could include: PMID35762751, PMID39178317, PMID37463183, PMID30087402, PMID31009518.

Reviewer #3: 1. Line 117: The authors employed normalized UIS counts, which is a commonly used method for identifying essential genes. However, considering that transposon insertions may exhibit certain positional biases, is there a risk of misclassification, such as false positives or false negatives? To my understanding, essential genes are typically defined as those whose knockout results in lethality. Therefore, directly defining genes identified through this non-targeted knockout approach as essential genes may not be entirely appropriate.

2. The authors identified 94 essential genes important for the in vivo adaptability of M1404 through mouse infection experiments. I am interested in understanding the rationale for selecting alsT_1, crp, and cyaA for the construction of individual mutant strains for further investigation. Since many other candidate genes were also identified, were targeted mutants of these genes constructed and analyzed as well? Did mutants of other genes exhibit similar infection phenotypes in mice, or did them fail to display significant phenotypic changes following screening? The authors focused their subsequent research on the crp and cyaA genes after gene screening. However, the function of the crp gene in Pasteurella multocida has already been reported in a previous studies (Identification of the crp gene in avian Pasteurella multocida and evaluation of the effects of crp deletion on its phenotype, virulence and immunogenicity. DOI: 10.1186/s12866-016-0739-y), and it is reasonable that cyaA, as an upstream gene, would affect the crp pathway. Therefore, conducting knockout studies on other screened genes that have not yet been reported may have greater scientific value.

3. Line 267: The authors state that inactivation of cpdA only slightly reduces the adaptability of VP161, yet targeted deletion of cpdA in VP161 could not be achieved, which appears somewhat contradictory. According to the Ecocyc database, cpdA is considered a non-essential gene in Escherichia coli, although the situation may differ in Pasteurella.

4. Lines 461–472: The authors constructed the corresponding mutants using a method similar to targetron and provided detailed tables of strains and primers. However, the manuscript lacks detailed descriptions of the procedures for vector construction. If possible, could the authors provide comprehensive methods for vector and mutant construction in the supplementary materials

5. I am curious about how the capsule was quantified in this study. In line 542, please specify the measurement method used, including the equipment (e.g., enzyme-linked immunosorbent assay (ELISA) reader, spectrophotometer), as well as the approach for quantifying hyaluronic acid. Were commercial kits, ELISA, or mass spectrometry used? As far as I know, polysaccharides are generally analyzed by SDS-PAGE silver staining or mass spectrometry. Please provide detailed experimental methods for capsule quantification as used in this study, as I could not find the relevant methodology in reference 7.

**Part III – Minor Issues: Editorial and Data Presentation Modifications**

Reviewer #1: It would be helpful if some of the key common genes identified by TraDIS in both strains are labelled in Fig 2A. I accept this will be limited, but it would be helpful to readers.

The animal ethics should be a stand-alone section.

Reviewer #2: Minor Concerns:

1. The authors state the P. multocida strains cause infection across many different types of animals (from mammals to birds). Can the authors add a brief sentence or two about how the infection manifests in the wild (symptomatically) vs. what is observed in the mouse model? This will help to contextualize the model for those not in the field since the affected species vary so widely.

2. To follow up Figure 1B, it would be helpful if the authors included a Venn diagram/Donut Chart/Heat Map or some other type of display that showed gene types which were distinct or conserved across the VP161 and M1404 strains. It is listed in the text (Lines 123-131) but may help the readers to see these displayed next to one another with relative differences in the number of genes belonging to each category.

3. While the Venn diagrams in Figure 1B, 2B-C are nice display items because they demonstrate proportionality of overlap, they are quite difficult to read. Could the authors modify colors so that distinct and overlapping sections are more obvious? Perhaps the authors could also display the Venn diagram data as a table?

Reviewer #3: 1. Line 172, The authors described the identification of dozens of genes associated with bacterial in vivo adaptability. Ultimately, the deletion of seven genes was found to enhance the strain’s in vivo adaptability. Does this mean that the presence of these genes is beneficial for adaptability, or rather, that their deletion improves adaptability? Please confirm whether this interpretation of the conclusion is correct.

2. In lines 175–199, the authors describe that multiple lipopolysaccharide or capsule synthesis-related genes are important for the in vivo adaptability of strains, mentioning many genes involved in polysaccharide synthesis. My question is whether all the genes mentioned have been reported in previous studies, or if they are novel discoveries made by this research group. If existing reports are involved, appropriate citations should be provided. For example, line 208 includes a relevant citation [28].

PLOS authors have the option to publish the peer review history of their article (what does this mean? ). If published, this will include your full peer review and any attached files.

**Do you want your identity to be public for this peer review?** For information about this choice, including consent withdrawal, please see our Privacy Policy .

Reviewer #1: No

Reviewer #2: No

Reviewer #3: **Yes: ** Saixiang Feng

**Figure resubmission:**

**Reproducibility:**



---

## [Decision Letter · Decision Letter 1]

19 Nov 2025

PPATHOGENS-D-25-01786R1

Comparative analysis of shared and unique mechanisms important for diverse strains of Pasteurella multocida to cause systemic infection in mice

PLOS Pathogens

Dear Dr. Boyce,

Thank you for submitting your manuscript to PLOS Pathogens. The three previous reviewers have all re-reviewed the revised version, and were overall very positive about it. One reviewer, however, indicates that the comments raised were addressed in the rebuttal letter, but had not made it into the manuscript. I do agree with this assessment and would like to ask you to carefully address this in a minor revision. The peer review is intended to improve the manuscript primarily for the readers, not for the reviewers themselves (or the editors), so clearly all points raised should be addressed in the rebuttal letter AND in the paper. Therefore, we invite you to submit a revised version of the manuscript that addresses the points raised during the review process.

We look forward to receiving your revised manuscript.

Kind regards,

Sebastian Suerbaum, MD

Academic Editor

PLOS Pathogens

David Skurnik

Section Editor

PLOS Pathogens

Sumita Bhaduri-McIntosh

Editor-in-Chief

PLOS Pathogens

orcid.org/0000-0003-2946-9497

Michael Malim

Editor-in-Chief

PLOS Pathogens

orcid.org/0000-0002-7699-2064

**Additional Editor Comments:**

Please incorporate in the main text your answers to the reviewers when feasible (see comment from Reviewer 2)

**Journal Requirements:**

1) Thank you for stating 'The P. multocida strain M1404 genome is available at Genbank: Biosample Accession Number SAMN49532731. All TraDIS data analysis output files are available at the NCBI Gene Expression Omnibus: Accession Number GSE300598.' However, we were unable to retrieve the data using the provided accession numbers. Please note that, though access restrictions are acceptable now, your entire minimal dataset will need to be made freely accessible if your manuscript is accepted for publication. This policy applies to all data except where public deposition would breach compliance with the protocol approved by your research ethics board. If you are unable to adhere to our open data policy, please kindly revise your statement to explain your reasoning and we will seek the editor's input on an exemption.

**Reviewers' Comments:**

Reviewer's Responses to Questions

**Part I - Summary**

Reviewer #1: Overall, I believe the manuscript has been fairly and thoroughly evaluated. The authors have thoughtfully and respectfully responded to the minor concerns raised in my initial review. In my opinion, they have also addressed the insightful comments raised by the other reviewers. In doing so, the authors have generated a body of work that will broadly impact the field. I commend the authors on an excellent study.

Reviewer #2: I remain enthusiastic about this manuscript and believe the authors have mostly address my original concerns. Some of my initial concerns are addressed in the Comments to Reviewers, but did not make it into the final manuscript and the authors cite being mindful of the final length of the manuscript/brevity as this reason. While this is understandable, I will defer to the editor on whether article length is a concern in this study or whether the authors should incorporate these comments into the final paper.

Reviewer #3: (No Response)

**Part II – Major Issues: Key Experiments Required for Acceptance**

Reviewer #1: No major issues.

Reviewer #2: (No Response)

Reviewer #3: (No Response)

**Part III – Minor Issues: Editorial and Data Presentation Modifications**

Reviewer #1: No minor issues.

Reviewer #2: (No Response)

Reviewer #3: (No Response)

PLOS authors have the option to publish the peer review history of their article (what does this mean? ). If published, this will include your full peer review and any attached files.

**Do you want your identity to be public for this peer review?** For information about this choice, including consent withdrawal, please see our Privacy Policy .

Reviewer #1: **Yes: ** Mark Schembri

Reviewer #2: No

Reviewer #3: No

**Figure resubmission:**
---

## [Editor Report · Decision Letter 2]

12 Dec 2025

Dear Dr Boyce,

We are pleased to inform you that your manuscript 'Comparative analysis of shared and unique mechanisms important for diverse strains of Pasteurella multocida to cause systemic infection in mice' has been provisionally accepted for publication in PLOS Pathogens.

Best regards,

Sebastian Suerbaum, MD

Academic Editor

PLOS Pathogens

David Skurnik

Section Editor

PLOS Pathogens

Sumita Bhaduri-McIntosh

Editor-in-Chief

PLOS Pathogens

orcid.org/0000-0003-2946-9497

Michael Malim

Editor-in-Chief

PLOS Pathogens

orcid.org/0000-0002-7699-2064
---

## [Editor Report · Acceptance letter]

Dear Dr Boyce,

We are delighted to inform you that your manuscript, "Comparative analysis of shared and unique mechanisms important for diverse strains of Pasteurella multocida to cause systemic infection in mice," has been formally accepted for publication in PLOS Pathogens.

Best regards,

Sumita Bhaduri-McIntosh

Editor-in-Chief

PLOS Pathogens

orcid.org/0000-0003-2946-9497

Michael Malim

Editor-in-Chief

PLOS Pathogens

orcid.org/0000-0002-7699-2064